# Genome-Wide Characterization and Expression Profiling of ABA Biosynthesis Genes in a Desert Moss *Syntrichia caninervis*

**DOI:** 10.3390/plants12051114

**Published:** 2023-03-01

**Authors:** Xiujin Liu, Xiaoshuang Li, Honglan Yang, Ruirui Yang, Daoyuan Zhang

**Affiliations:** 1State Key Laboratory of Desert and Oasis Ecology, Key Laboratory of Ecological Safety and Sustainable Development in Arid Lands, Xinjiang Institute of Ecology and Geography, Chinese Academy of Sciences, Urumqi 830011, China; 2Xinjiang Key Lab of Conservation and Utilization of Plant Gene Resources, Xinjiang Institute of Ecology and Geography, Chinese Academy of Sciences, Urumqi 830011, China; 3University of Chinese Academy of Sciences, Beijing 100049, China; 4Turpan Eremophytes Botanical Garden, Chinese Academy of Sciences, Turpan 838008, China

**Keywords:** ABA biosynthesis, evolution, bryophyte, chlorophyte, desiccation tolerance

## Abstract

*Syntrichia caninervis* can survive under 80–90% protoplasmic water losses, and it is a model plant in desiccation tolerance research. A previous study has revealed that *S. caninervis* would accumulate ABA under dehydration stress, while the ABA biosynthesis genes in *S. caninervis* are still unknown. This study identified one *ScABA1*, two *ScABA4s*, five *ScNCEDs*, twenty-nine *ScABA2s*, one *ScABA3*, and four *ScAAOs* genes, indicating that the ABA biosynthesis genes were complete in *S. caninervis*. Gene location analysis showed that the ABA biosynthesis genes were evenly distributed in chromosomes but were not allocated to sex chromosomes. Collinear analysis revealed that *ScABA1*, *ScNCED*, and *ScABA2* had homologous genes in *Physcomitrella patens*. RT-qPCR detection found that all of the ABA biosynthesis genes responded to abiotic stress; it further indicated that ABA plays an important role in *S. caninervis*. Moreover, the ABA biosynthesis genes in 19 representative plants were compared to study their phylogenetic and conserved motifs; the results suggested that the ABA biosynthesis genes were closely associated with plant taxa, but these genes had the same conserved domain in each plant. In contrast, there is a huge variation in the exon number between different plant taxa; it revealed that ABA biosynthesis gene structures are closely related to plant taxa. Above all, this study provides strong evidence demonstrating that ABA biosynthesis genes were conserved in the plant kingdom and deepens our understanding of the evolution of the phytohormone ABA.

## 1. Introduction

Abscisic acid (ABA) is a very important carotenoid-derived phytohormone and plays a significant role in plant growth and development under biotic or abiotic stress conditions [1]. As a sesquiterpene structure compound (C_15_), ABA is synthesized directly in fungi [2,3], while in plants, it is usually produced through carotenoid cleavage (C_40_), which is an indirect synthetic pathway (Figure 1A) [4,5]. The first step in the ABA biosynthetic pathway is the conversion of zeaxanthin to all-*trans*-violaxanthin by zeaxanthin epoxidase (ZEP) [6]. *ABA1* encodes ZEP in *Arabidopsis thaliana* [7]. All-*trans*-violaxanthin is then converted to 9′-*cis*-neoxanthin and 9′-*cis*-violaxanthin, which are, respectively, catalyzed by neoxanthin synthase (NSY) and an unknown isomerase [8]. *ABA4* encodes NSY in *A. thaliana* [9]. The next step, 9′-*cis*-violaxanthin and 9′-*cis*-neoxanthin cleavage by a 9-*cis*-epoxycarotenoid dioxygenase (NCED) is generally considered the rate-limiting step in the ABA biosynthetic pathway [10]. The cleavage product xanthoxin is then converted to abscisic aldehyde by a short-chain alcohol dehydrogenase encoded by the *ABA2* gene [11]. The final step, abscisic aldehyde oxidizes to ABA, is catalyzed by an abscisic aldehyde’s oxidation (AAO) [12]. *Arabidopsis ABA3* encodes molybdenum cofactor (MoCo), which is required for AAO catalytic activity and ABA synthesis [13,14]. Functionally, *ABA1*, *ABA4*, *NCED*, *ABA2*, *ABA3*, and *AAO* are key genes in ABA biosynthetic pathway.

Expression of ABA biosynthesis genes is closely related to endogenous ABA levels, so they will directly determine ABA’s action in different plant stress responses. Several studies have investigated the evolution of ABA biosynthesis-related genes across land plant lineages using bioinformatics, but very few are known outside of angiosperms, including species of gymnosperms, ferns, lycophytes, bryophytes, and chlorophytes; thus, this is an expanding field [15]. This research has revealed novel insights into the ABA biosynthesis pathway across different plant taxa, especially in bryophytes. Bryophytes are key organisms in plant evolution, and clarification of their ABA biosynthesis processes is important for understanding land plant evolutionary adaptation [16]. The moss *Physcomitrella patens* is the model plant in bryophyte research [17]; recent research on the *P. patens* ABA biosynthesis pathway has revealed it contain *PpABA1*, *PpABA4*, *PpNCED*, and *PpABA3* genes and they play roles in plant stress response. For example, *PpABA1* can enhance osmotic acclimation, and the transcript abundance of *PpNCED* is significantly increased under salt stress, while *PpABA2* and *PpAAO* have not been reported [18,19,20]. In general, the data on bryophytes’ ABA biosynthesis are currently scarce.

One of the most striking aspects of ABA biosynthesis is the drastic increase in ABA levels during dehydration [21], and an ABA-induced increase in stress tolerance is also reported in other land plant lineages, including nonvascular bryophytes that diverged from vascular plants more than 480 million years ago. Desiccation tolerance (DT) moss is a type of bryophyte that can survive under 80–90% protoplasmic water loss [22]. ABA plays an important role in DT moss, as in angiosperms, ferns, lycophytes, and other desiccation-tolerant plants [21]. *Syntrichia caninervis* Mitt. is a desert moss distributed in the arid land of Central Asia and North America [23,24]; it is one of the dominant species in the biological soil crusts of deserts, and it plays an important role in ecological restoration and biodiversity conservation in the desert ecosystem [25]. *S. caninervis* is the only DT moss for which its genome has been sequenced [26]; it propagates rapidly and can regenerate a large number of gametophytes within one month [27]. Moreover, *S. caninervis* contains a large number of resistance genes, such as *ScELIPs*, *ScALDHs*, and *ScDREBs* [28,29,30,31]. From the above discussion, it appears that *S. caninervis* is not only a model plant for the study of DT mechanism and excellent stress tolerance gene mining [32] but also a typical material to study the acquisition of the ABA biosynthesis genes.

As reported in the literature, *S. caninervis* is capable of ABA biosynthesis, and ABA concentrations increase under dehydration stress [32,33]. Nevertheless, until now, no data on *S. caninervis* ABA biosynthesis genes have been reported. In this study, we identified and analyzed one *ScABA1* gene, two *ScABA4* genes, five *ScNCED* genes, twenty-nine *ScABA2* genes, one *ScABA3* gene, and four *ScAAO* genes in the *S. caninervis* genome, and investigated their expression pattern under different abiotic stresses. In addition, we also identified and compared ABA biosynthesis genes in additional bryophytes, chlorophytes, and other plant taxa. The present study, therefore, focused on the ABA biosynthesis genes in early-diverging plant species, including bryophytes and chlorophytes, and recent progress in the model desiccation-tolerant moss *S. caninervis* and the model moss *P. patens*. With recent advances in our capacity to characterize gene information in different plant taxa, we are on the cusp of revealing the origins of these critical ABA biosynthesis genes and understanding how ABA may have shaped land plant adaptation.

## 2. Results

### 2.1. Identification and Characteristic of ABA Biosynthesis Genes in S. caninervis

Combining the results of BLASTP and HMMER, a total of 42 ABA biosynthesis genes were identified in the *S. caninervis* genome, including one *ScABA1* gene, two *ScABA4* genes, five *ScNCED* genes, twenty-nine *ScABA2* genes, one *ScABA3* gene and four *ScAAO* genes (Table 1 and Appendix A). ScABA1 protein sequence analysis showed that it encoded 681 amino acids (aa) with an average molecular weight (MW) at 73.9 kDa and isoelectric points (pI) at 8.15. Simultaneously, subcellular mapping prediction indicated that the *ScABA1* gene was located in the chloroplast. The predicted protein lengths of *ScABA4* genes (*Sc_g04273*/*Sc_g01818*) were 252 and 199 amino acids; ScABA4 proteins’ molecular weights were 27.8 and 13.6 kDa, pI were 9.71 and 7.66, and they were located in chloroplasts and the endomembrane, respectively. The length of ScNCED proteins ranged from 540 to 615 amino acids, the predicted molecular weight ranged from 60.5 to 68.2 kDa, and pI ranged from 5.59 to 7.57, and these *ScNCED* genes were located in chloroplasts and cytoplasm. As *S. caninervis* had a number of *ScABA2* genes, we chose the top five *ScABA2* genes according to the E-values of the BLASTP screen for further comparison and analysis of their characteristics. The ScABA2 protein sequences ranged from 254 to 335 amino acids in length, their molecular weights ranged from 26.1 to 36.7 kDa, their pI ranged from 5.52 to 8.24, and most of them were located in the mitochondrion. ScABA3 protein sequence analysis showed that it was composed of 919 amino acids, its molecular weight was 101.8 kDa, pI was 7.1, and it was located in the extracellular space. Four ScAAO protein sequences analysis showed that their lengths ranged from 1273 to 1334 amino acids, molecular weights ranged from 138.5 to 144.9 kDa, pI ranged from 6.14 to 6.67, and most of them were located in the cytoplasm.

To comprehensively evaluate the characteristics of the ABA biosynthesis genes in *S. caninervis*, we also identified these genes in the model bryophyte *P. patens* genome. The result indicated that the ABA biosynthesis gene number in *P. patens* was the same as that in *S. caninervis*, except that *P. patens* had eight *PpNCEDs* and twenty *PpABA2s* (Table 1 and Appendix A). In total, the characteristics of ABA biosynthesis genes were similar in both bryophytes, which suggested that these genes were conserved in bryophyte species. The PpABA1 protein was 685 amino acids in length, with a predicted molecular mass of 74.4 kDa and pI of 7.85, and it was located in chloroplast too. The predicted protein lengths of *PpABA4* genes (*Pp3c12_21540*/*Pp3c3_37490*) were 297 and 264 amino acids, PpABA4 proteins molecular weights were 33 and 29.2 kDa, pI were 10.03 and 9.69, and they were located in mitochondrion and chloroplast respectively. All PpNCED proteins ranged from 538 to 622 amino acids in length, molecular weights ranged from 61.2 to 69.9 kDa, pI ranged from 5.63 to 5.84, and they were mainly located in the cytoplasm. PpABA2 protein sequence analysis showed that they encoded 306–359 amino acids, molecular weight ranged from 31.7 to 39.3 kDa, putative pI ranged from 5.89 to 8.77, and they were dispersedly localized in the chloroplast, mitochondrion, and cytoplasm. PpABA3 protein sequence analysis showed that it encoded 940 amino acids, the molecular weight was 104.2 kDa, and the pI was 6.34. PpABA3 was located in chloroplast, which distinguished it from *S. caninervis*. PpAAO protein sequences analysis showed that they were 1283 to 1396 amino acids in length, predicted molecular mass ranged from 139.2 to 152.2 kDa, pI ranged from 5.86 to 6.46, and they were mainly located in the cytoplasm.

### 2.2. Gene Number of ABA Biosynthesis Genes in Different Plant Taxa

On the basis of the identification and characterization of the ABA biosynthesis genes in *S. caninervis* and *P. patens*, it can be concluded that the ABA biosynthesis genes existed in bryophytes. Nevertheless, the comprehensive analysis of ABA biosynthesis genes in chlorophyte species has not been reported. To better understand the evolution of the ABA biosynthesis genes, here, we summarized the number of *ABA1*, *ABA4*, *NCED*, *ABA2*, *ABA3*, and *AAO* genes among 19 representative species from chlorophytes to angiosperms (Figure 1). As shown in Figure 1B, all of the ABA biosynthesis genes were present in different plant taxa, whether in lower or higher plants.

Specifically, *ABA1* is a single gene in all plant species except *Z. mays* and *P. trichocarpa* (Figure 1B), so the *ABA1* gene quantity is very stable in different plant genomes. The result of the *ABA4* gene number analysis indicated that the bryophyte was a transition taxon because all of the lower plant chlorophytes had two *ABA4* genes, and most of the higher plant angiosperm had only one *ABA4* gene; however, some bryophytes had one, and the other bryophytes had two *ABA4* genes, such as *P. patens*, *S. caninervis*, and *C. purpureus* (Figure 1B). According to the identification of the *NCED* genes in 19 plants, we found that all plant species had five *NCED* genes at least; angiosperms had the largest number of *NCED* genes, ranging from 5 to 26; chlorophytes had the lowest number of *NCED*, ranging from 5 to 7, and bryophyte plants were in the middle, ranging from 5 to 12 (Figure 1B). The number of *ABA2* genes suggested that bryophyte was in the middle ranging from 17 to 29, with the minimum in chlorophyte ranging from 4 to 11 and the maximum in angiosperms ranging from 12 to 56 (Figure 1B). *ABA2* belongs to the short-chain dehydrogenase/reductase (SDR) family; there are 56 *ABA2*/*SDR* genes in *A. thaliana*, but only one *AtABA2* (*AT1G52340*) participates in ABA biosynthesis. Moreover, each plant had many *ABA2* genes in our results, but we can not accurately confirm which ones are involved in ABA biosynthesis merely by using bioinformatic analysis. As *ABA3* is a single gene, its number in 19 plants was similar to that of the *ABA1* gene (Figure 1B). Each chlorophyte had only one *AAO*, which differed from bryophytes (3–7 *AAO* genes) and angiosperms (2–6 *AAO* genes). All in all, the number of ABA biosynthesis genes was closely related to plant taxa.

### 2.3. Phylogenetic and Conserved Motif Analysis of ABA Biosynthesis Genes in Various Plants

Great progress regarding ABA biosynthesis genes has been made in angiosperms; however, little attention has been given to these genes in early-diverging plant species such as bryophytes and chlorophytes. To better understand the evolutionary relationships within all the *ABA1*, *ABA4*, *NCED*, *ABA2*, *ABA3*, and *AAO* genes in different plant taxa, phylogenetic and conserved motif analysis was conducted.

A maximum likelihood phylogenetic tree was constructed to explore the evolutionary relationship of 21 *ABA1* genes from 19 species inferred from the amino acid sequences in IQTree v1.6.12 using the best-fit substitution model (WAG + I + G_4_), which was automatically selected by the software according to the Bayesian information criterion scores and weights (BIC) with partitions. As shown in the phylogenetic tree (Figure 2A), the ABA1 protein sequences of *S. caninervis*, left bryophyte, lycophyte, and fern species were clustered together in one clade, which means that bryophyte species were closely related to each other. In addition, the ABA1 of three chlorophyte species and gymnosperm were clustered together in one clade, and angiosperm species were clustered together. The results of the conserved motif analysis showed that the ScABA1 protein was composed of the FAD_binding_3 and FHA domain, which was consistent in bryophytes even in different plant taxa (Figure 2B). The *A. trichopoda* ABA1 protein is an exception because it did not contain the FHA domain. Moreover, most of the ABA1 proteins had two FAD_binding_3 domains, while a few of them had only one FAD_binding_3 domain, such as *Z. mays*, *C. richardii*, and *S. fallax*. Consequently, the phylogenetic and conserved domain results indicated that *ABA1* is highly conserved in different plants.

An ML phylogenetic tree of 28 *ABA4* genes from 19 plant species was constructed in IQTree using the best-fit substitution model (JTT + F + I + G_4_), which was automatically selected by the software. As with ScABA1, ScABA4 was clustered together with other bryophyte, fern, and lycophyte species; among them, *S. caninervis* was closely associated with *C. purpureus*, *B. argenteum*, and *P. patens*. All three chlorophyte species, including six ABA4 proteins, formed one clade and had a close relationship. Moreover, the *ABA4* genes of angiosperm species were clustered together and formed three clades. Meanwhile, a separate clade was formed for *T. plicata* (Thupl.29379874s0002), a gymnosperm, and another separate clade was formed for *A. thaliana* (AT1G67080) in the angiosperm ABA1 clade (Figure 3A). In line with previous studies, ABA4 contains a DUF4281 domain, which is widely distributed across all plant taxa (Figure 3B). At the same time, we found that there were shorter protein lengths in *S. caninervis* and *S. moellendorffii*. Above all, we demonstrated that *ABA4* is highly conserved in the different plant taxa.

In fact, we identified 206 *NCED* genes from 19 plant species, and the ML tree was constructed from the top five *NCED* genes of each species based on the E-value of BLASTP result in IQTree using the best-fit substitution model (LG + F + R_9_). We found that *ScNCED* genes in *S. caninervis* were always clustered together with *C. purpureus*; simultaneously, *ScNCEDs* had a close relationship with other bryophyte species. In addition, three chlorophyte species, including 15 NCED proteins, formed two clades, NCED proteins of fern, lycophyte, and gymnosperm were respectively clustered together, NCED proteins of angiosperm were scattered in the phylogenetic tree (Appendix A). The conserved motif analysis showed that all NCED proteins had one RPE65 domain, which was widely distributed across all plant taxa (Appendix A). In sum, the *NCED* gene is highly conserved, too.

*ABA2* belongs to a multigene family, and we identified 471 *ABA2* genes from 19 species. The *ABA2* evolutionary tree was constructed from the top five *ABA2* genes of each species as an *NCED* phylogenetic tree, and LG + R_5_ was the best-fit substitution model in IQTree for the construction of the ML phylogenetic tree of ABA2. The result showed that the classification of the phylogenetic tree was ambiguous because it did not cluster by plant taxa (Appendix A). Meanwhile, *ScABA2* genes were always clustered together with different bryophyte species. Our conserved motif analysis showed that each ABA2 protein had an adh_short_C2, adh_short_C, and KR domain (Appendix A). In accordance with the phylogenetic and conserved domain results, *ABA2* is very conserved in different plants.

An ML phylogenetic tree based on 20 *ABA3* genes was constructed in IQTree using the best-fit substitution model (JTT + I + G_4_), which was automatically selected by the software. As shown in Figure 4A, *ScABA3* was sister clade to *C. purpureus*, *B. argenteum*, *P. patens* and *S. fallax* and had a close relation. The ABA3 proteins of three chlorophytes, *S. moellendorffii*, and *M. polymorpha*, clustered together in one clade, and they were closely related to each other. The *ABA3* of angiosperm species were clustered together; a separate clade was formed for *T. plicata* (Thupl.29378725s0003), a gymnosperm, and another separate clade was formed for *A. trichopoda* (AmTr v1.0 scaffold00032), an angiosperm (Figure 4A). The conserved motif analysis showed that ABA3 contained one MOSC, MOSC_N, Thioredoxin domain, and two Aminotran_5 domains in each plant taxon, except for *B. argenteum*, *S. moellendorfii*, and *C. reinhardtii* (Figure 4B). Therefore, *ABA3* is very conserved in the different plant taxa.

Finally, sixty-three *AAO* genes were identified for ML phylogenetic tree construction in IQTree using the best-fit substitution model (LG + F + R_6_). The result showed that the phylogenetic tree of *AAO* genes was different from other ABA biosynthesis genes because they did not cluster by plant taxa. However, we found that ScAAO proteins were always clustered together with *C. purpureus* AAO proteins, which were similar to other ABA biosynthesis genes (Figure 5A). The conserved motif analysis suggested that each AAO protein had Fer2, Fer2_2, FAD_binding_5, Ald_Xan_dh_C, MocoBD1, MocoBD2, and CO_deh_flav_C domains (Figure 5B). In consequence, the phylogenetic and conserved domain results confirmed that AAO is very conserved in the different plant taxa.

### 2.4. Structural Characterization of ABA Biosynthesis Genes in Various Plants

To discover the differences in ABA biosynthesis genes, we analyzed the structural characteristics of these genes among different plant taxa (Appendix A) and compared the exon numbers in 19 plant species from different plant taxa (Figure 6). Through analysis of the *ABA1* gene structure, it was found that *ScABA1* had two exons; unexpectedly, the *ABA1* exon number in bryophytes had a great difference when compared with other plant taxa, as most of them only had one or two exons, such as *S. caninervis*, while *M. polymorpha* and *S. fallax* had 16 exons. In addition, gymnosperm, fern, lycophyte, and the majority of angiosperm *ABA1* genes had 16 exons. *A. trichopoda ABA1* had 27 exons which was the most numerous, while *P. patens* and *C. purpureus* only had one exon, which were the least numerous. The *ABA1* genes from three chlorophytes had 8–10 exons. Thus, the gene structure of *ABA1* exhibited extreme variation (Figure 6 and Appendix A). Through analysis of the *ABA4* gene structure, we found that the majority of angiosperm species had six exons, bryophyte species had seven exons, and chlorophytes had 4–6 exons (Appendix A). Overall, the variation in the *ABA4* exon number was not obvious (Figure 6). *NCED* is a large gene family; we retained the top five *NCED* genes according to the E-value to further analyze the gene structure. Bryophyte *NCED* gene structures were different from other taxa because these genes differed greatly in their exon number. Some *NCED* genes had one or less than five exons; the others had 13–15 exons (Appendix A). Taking *S. caninervis*, for example, the exon number of *ScNCED* genes was one (*Sc_g07550*), four (*Sc_g09383*), fourteen (*Sc_g14527*), fourteen (*Sc_g11853*), and thirteen (*Sc_g07528*), which shows a great variation. However, a similar gene structure was predicted between angiosperm species *NCED* genes composed by one exon, except that *Zm00001d031086* had two exons in *Z. mays*; *LOC_Os12g44310* had fourteen exons in *O. sativa*; *Zosma01g05390* and *Zosma01g01740* had two and fifteen exons in *Z. marina*; and *AmTr_v1.0_scaffold00022* and *AmTr_v1.0_scaffold00056* had fourteen and nine exons in *A. trichopoda*. At the same time, the *NCED* exon-intron structures of gymnosperm, fern, and lycophyte species were similar to those in angiosperm species. Distinctly, each chlorophyte *NCED* gene contained more than 10 exons (Appendix A). Hence, *NCED* gene structures were varied in the different evolutionary branches (Figure 6). Through analysis of the *ABA2* gene structure, it was found that the *ScABA2* exon number ranged from 1 to 6, and it was the same in each bryophyte. Most angiosperm, gymnosperm, fern, and lycophyte species contained two exons; a few of them differed, such as *Zm00001d049277*, which had three exons, while *Potri.016G073800* had only one, *Zosma03g33130* had eleven exons, and so on. The exon number of chlorophyte species ranged from 7 to 12 exons; consequently, chlorophytes had many more exons (Figure 6 and Appendix A). In terms of the prediction result of the *ABA3* gene structure, we found that the exon number of *ABA3* genes was stabilized at 20–23 exons, with the exception of three chlorophyte species that contained 8–16 exons, which were obviously lower than in other plants (Figure 6 and Appendix A). Based on the *AAO* gene structure, we revealed that the exon number showed a great difference in *S. caninervis*, ranging from 7 to 27, with a similar result in bryophytes. However, the majority of angiosperm, gymnosperm, fern, and lycophyte *AAO* genes had 10 exons. Moreover, the *AAO* genes of two chlorophyte species had thirty-two exons, which was the largest number, while *Zm00001d019376* only had four exons, which was the lowest (Appendix A). The exon number of *AAOs* had a great difference in each evolutionary branch, with huge variation (Figure 6). In conclusion, the results further indicated that the ABA biosynthesis gene structures were associated with plant taxa.

### 2.5. Chromosomal Locations and Collinearity Analysis of ABA Biosynthesis Genes in S. caninervis and P. patens

In order to understand the distribution traits of ABA biosynthesis genes, we chose *S. caninervis* [26] and another model moss, *P. patens* [34], as examples to evaluate the ABA biosynthesis gene locations in the chromosomes (Figure 7). In *S. caninervis*, a desiccation-tolerant moss, *ScABA1* and *ScABA3* are each a single gene, both located on Chromosomes 10. In addition, *ScABA4* genes are distributed on Chromosomes 10 and 12; *ScNCED* genes are distributed on Chromosomes 2, 5, 7, and 8; *ScAAO* genes are distributed on Chromosomes 1, 3, 6, and 11; and *ScABA2* genes are distributed on every chromosome except for Chomosomes 8 and 13. In *P. patens*, a model moss plant, *PpABA1* and *PpABA3* are also each a single gene, distributed on Chromosomes 12 and 5, respectively; *PpABA4* genes are distributed on Chromosomes 3 and 12; *PpNCED* genes are distributed on Chromosomes 6, 12, 16, 18, 21, 22 and 25; and *PpAAO* genes are distributed on Chromosomes 2, 5, 19 and 20. The *PpABA2* gene is a huge gene family, so it is distributed among half of all chromosomes.

Given that tandem duplicates play an important role in the creation of gene families, and *NCED*, *ABA2*, and *AAO* belong to a multigene family, we examined the tandem repeat clustering of ABA biosynthesis genes in two bryophyte genomes. Based on the results of ABA biosynthesis gene location, we found that two *ScABA2* tandem repeat clusters (*Sc_g14027*/*Sc_g14030*, *Sc_g03294*/*Sc_g03296*) were located on Chromosome 3 and 11, one *PpNCED* tandem repeat cluster (*Pp3c25_4816*/*Pp3c25_4810*) and one *PpABA2* tandem repeat cluster (*Pp3c18_20230*/*Pp3c18_20320*) located on Chromosomes 25 and 18 (Figure 7). However, no tandem repeat cluster was found for *AAO* gene families; these genes were located on different chromosomes in both bryophytes. In conclusion, the ABA biosynthesis genes were evenly distributed and dispersed on the chromosomes of *S. caninervis* and *P. patens*. Furthermore, Chromosomes 27 and 13 are sex chromosomes in *S. caninervis* and *P. patens*, respectively, which similarly did not host the ABA biosynthesis genes. There was a high similarity in the distribution of ABA biosynthesis genes between *S. caninervis* and *P. patens*. Therefore, we further analyzed the collinearity of ABA biosynthesis genes in the genomes of the two bryophyte species.

Comparison of the ABA biosynthesis genes between *S. caninervis* and *P. patens* can provide insights into the evolutionary history of ABA (Figure 8). It was found that nine ABA biosynthesis genes were homologous between the two bryophytes. In *S. caninervis*, the *ScABA1* gene can be found; as in *P. patens*, *ABA1* as a single gene has a significantly collinear relationship among bryophyte species. Among six *NCED* genes in the *S. caninervis* genome, only one *ScNCED* (*Sc_g14527*) gene has homologous genes in *P. patens*. We found that seven *ABA2* genes had homologous genes between *S. caninervis* and *P. patens*. This result suggested that *ABA1*, *NCED*, and *ABA2* genes may be generated from the same ancestral gene and have the same gene function. However, the *ABA4*, *ABA3*, and *AAO* genes of the two bryophyte species were not homologous, and it was speculated that these genes are more likely to be an independent event in the evolution of bryophytes.

### 2.6. Expression Analysis of ABA Biosynthesis Genes in S. caninervis under Different Stresses

ABA plays a key role in many types of abiotic stress in plants. Based on the information analysis results, we detected the expression patterns of ABA biosynthesis genes in the resistance model moss *S. caninervis* under abiotic stress. It is very important to analyze the expression of ABA biosynthesis genes under dehydration, salt, cold, heat, and ABA stress (Figure 9). Under dehydration stress, the expression of *ScABA1* rapidly increased approximately 20-fold and was sustained, except for 6 h and 24 h; *ScABA4* had a slight fluctuation within 2-fold, *ScNCED* was down-regulated, *ScABA2* slightly increased between 2- and 3-fold, *ScABA3* was significantly up-regulated at the early stage, and *ScAAO* rapidly increased 5–10-fold and then declined to low abundance at 24 h. Under salt stress, the *ScABA1* transcript abundance gradually accumulated to 38-fold before 2 h and decreased to 21–31-fold; *ScABA4* and *ScABA2* had a slight alteration within 2–3-fold, *ScNCED* was barely changed, *ScABA3* was expressed at the late stage at approximately 4–6-fold, *ScAAO* rapidly increased 9-fold in 1 h but subsequently declined. Under cold stress, *ScABA1* maintained high expression from 14- to 62-fold and peaked at 12 h; *ScABA4* sustained low expression and peaked at 24 h at approximately 5-fold; *ScNCED* was almost unchanged and showed negligible levels at 12 h; *ScABA2* was up-regulated and peaked at 12 h; *ScABA3* had a high expression at 1 h, 12 h, and 24 h, *ScAAO* was rapidly increased and sustained at 6–10-fold. Under heat stress, *ScABA1* was rapidly and constantly up-regulated from 16- to 30-fold; the transcript abundance of *ScABA4*, *ScABA2*, and *ScABA3* increased but not significantly; *ScNCED* significantly decreased; *ScAAO* rapidly increased to 8-fold and then slightly declined to 5–6-fold. Under ABA stress, *ScABA1* gradually increased to 8-fold at 12 h; the expression of *ScABA4*, *ScABA3*, and *ScAAO* was at negligible levels; *ScNCED* was up-regulated before 1 h but subsequently declined to low abundance, and *ScABA2* had high expression at 24 h at approximately 6-fold.

## 3. Discussion

With the continuous development of sequencing technology, the genomes of bryophytes and chlorophytes have been published rapidly [26,34,35,36], and these lay a solid foundation for further analysis and research on the origin of ABA in early-diverging plant species and also provide sufficient information for the study of the ABA biosynthetic pathway. In this study, we analyzed ABA biosynthesis genes and, for these genes, identified their evolutionary relationships, conserved domain, gene structure, chromosome location, collinearity, and expression pattern in *S. caninervis* and 18 representative plants from different plant taxa for the first time.

Previous studies have shown that *S. moellendorffii*, *P. patens*, and *C. reindardtii*, the model plant of lycophytes, bryophytes, and chlorophytes, respectively, do not contain *ABA2* and *AAO* genes, suggesting that *ABA2* and *AAO* appear only in angiosperms; thus, they speculated upon the presence of an alternative enzymatic pathway to convert xanthoxin to ABA [19]. In addition, little attention has been paid to the ABA biosynthetic pathway in chlorophytes; only *ABA1*, *ABA4*, and *ABA3* genes were previously reported [37]. On the whole, there is a poor understanding of the ABA biosynthetic pathway in early-divergence plants. Thanks to the release of more genomes, we first found that *S. caninervis* contained a complete set of ABA biosynthesis genes, and this was consistent in other bryophyte species (Figure 1B). Unexpectedly, three chlorophyte species had all of the ABA biosynthesis genes too. Similar to past research, our results showed that the *ABA1* and *ABA3* genes are a single gene in bryophytes and chlorophytes, as in angiosperms [7,14]. Compared with higher plants, chlorophyte plants had many more *ABA4* genes. On the contrary, the quantity of *NCED*, *ABA2*, and *AAO* genes was much lower in chlorophytes, which suggested that these genes perhaps increased during evolution (Figure 1B). In total, both lower and higher plants had ABA biosynthesis genes, and the number of ABA biosynthesis genes had a close relationship with the plant taxa; simultaneously, the results further indicated that the ABA biosynthetic pathway was conserved in the plant kingdom.

Previous reports revealed that the *ABA1*, *ABA4*, and *NCED* genes were conserved in angiosperm species by domain analysis [37,38]. In *S. caninervis*, we found that the conserved domain of the ABA biosynthesis genes was the same as in the rest of bryophytes, chlorophytes, and other plant taxa; for example, ScABA4 proteins also had the DUF4281 domain (Figure 3), and ScNCED proteins also had the RPE65 domain (Appendix A). Therefore, this result provided evidence that the domain of ABA biosynthesis genes was highly conserved in both lower and higher plants. On the basis of phylogenetic tree analysis, the single genes, such as *ABA1*, *ABA4*, and *ABA3*, were clearly clustered together by plant taxa (Figure 2, Figure 3 and Figure 4), while *NCED*, *ABA2*, and *AAO* belonged to a multigene family and their classification in the phylogenetic tree was ambiguous (Figure 5, Appendix A). Besides the AAO phylogenetic tree, ABA biosynthesis genes are closely associated with plant taxa.

In the highly conserved ABA biosynthesis genes, gene structure had unanticipatedly large changes, as indicated by the wide range of exon numbers between the plant taxa, with the exception of *ABA4* (Figure 6 and Appendix A). The exon number of *ABA1* and *ABA3* genes increased during plant evolution (Appendix A); conversely, *NCED*, *ABA2*, and *AAO* had far fewer exons in higher plants (Appendix A). In addition, bryophytes proved to be a turning point in the variation in exon number; for example, the majority of angiosperm, gymnosperm, fern, and lycophyte *NCED* genes were composed of one exon, and chlorophyte species had more than 10 exons; while some bryophyte species had one exon, the rest of the bryophytes had 13–15 exons (Appendix A). According to the ABA biosynthesis gene structure analysis, it can be concluded that bryophytes were the transitional form in plant evolution; in line with past research, bryophytes were the earliest diverging lineages of the extant land plants and the most original group of existing higher plants [39,40]. Consequently, the differences in ABA biosynthesis gene structures were affected by plant taxa.

On the basis of the analysis of the ABA biosynthesis genes distribution in *S. caninervis* and *P. patens*, these genes were evenly distributed and dispersed on the chromosomes, but they were not located on sex chromosomes in the two bryophytes (Figure 7). Through collinearity analysis, *ABA1*, *NCED*, and *ABA2* genes showed homologous genes between the *S. caninervis* and *P. patens* genomes (Figure 8). This provides new evidence that the ABA biosynthetic pathway originated from the same ancestral gene and shared the same stress resistance function.

When organisms start to colonize terrestrial habitats, endogenous ABA is increased even under mild drought stress; then, desiccation-protecting mechanisms are stimulated [15,41]. In this study, an extremely desiccation-tolerant moss was chosen as a representative to examine whether the ABA biosynthesis genes of bryophytes respond to abiotic stress. In *S. caninervis*, all ABA biosynthesis genes changed their expression under five types of treatments to cope with abiotic stress (Figure 9). Moreover, the expression of *ScABA1* concerning abiotic stress was more profound than that of other genes. However, it is worth noting that ABA synthesis genes had lower expression levels under 10 μM ABA than other treatments, except for the *ScNCED* and *ScABA2* genes. Based on the transcriptional expression profiles of the ABA biosynthesis genes under different stresses, ABA could play a key regulatory role in abiotic stress in desiccation tolerance moss, as shown in a previous study [33].

In this article, we focused on the ABA biosynthesis genes in early-diverging plants, especially bryophytes. Although the number and structure of ABA biosynthesis genes were affected by plant taxa, these genes existed and were intact in both lower and higher plants. Our results provided solid evidence that the ABA biosynthesis genes were conserved in the plant kingdom, which is critical in understanding the role of ABA in early-diverging plant species. On the other hand, ABA biosynthesis genes are involved in responding to all types of abiotic stress, which suggests that ABA plays an important role in responding to abiotic stress in bryophytes. While bryophytes such as *S. caninervis* gametophytes do not have a stoma, how does the ABA biosynthetic pathway participate in the stress resistance of bryophytes? The data are currently scarce, and more work is required to investigate ABA biosynthesis in bryophytes in the future.

## 4. Materials and Methods

### 4.1. Sequence Collection of ABA Biosynthesis Genes in Various Plant

Using the sequence of the *A. thaliana* ABA biosynthesis gene (*AtABA1*/*AT5G67030*, *AtABA4*/*AT1G67080*, *AtNCED3*/*AT3G14440*, *AtABA2*/*AT1G52340*, *AtABA3*/*AT1G16540*, *AtAAO3*/*AT2G27150*) as a query, the homologous sequences were retrieved from Phytozome (https://phytozome-next.jgi.doe.gov/ (accessed on 22 January 2023)) by using BLASTP with an E-value cutoff of 1e^−6^. A collection of ABA biosynthesis sequences entries was identified from 19 plant species comprising angiosperms: *Zea mays* (NCBI taxonomy ID 4577) [42], *Oryza sativa* (NCBI taxonomy ID 39947) [43], *Populus trichocarpa* (NCBI taxonomy ID 3694) [44], *A. thaliana* (NCBI taxonomy ID 3702) [45], *Vitis vinifera* (NCBI taxonomy ID 29760) [46], *Zoster marina* (NCBI taxonomy ID 29655) [47], *Amborella trichopoda* (NCBI taxonomy ID 13333) [48]; gymnosperm: *Thuja plicata* (NCBI taxonomy ID 3316) [49]; fern: *Ceratopteris richardii* (NCBI taxonomy ID 49495) [50]; lycophyte: *Selaginella moellendorffii* (NCBI taxonomy ID 88036) [51]; and bryophyte: *Marchantia polymorpha* (NCBI taxonomy ID 3197) [52], *Ceratodon purpureus* (NCBI taxonomy ID 3225) [35], *Sphagnum fallax* (NCBI taxonomy ID 53036) [53], *P. patens* (NCBI taxonomy ID 3218) [34], *S. caninervis* (NCBI taxonomy ID 200751) [26], *Bryum argenteum* (NCBI taxonomy ID 37413); Chlorophyte: *Chlamydomonas reinhardtii* (NCBI taxonomy ID 3055) [54], *Dunaliella salina* (NCBI taxonomy ID 3046) [55], and *Volvox carteri* (NCBI taxonomy ID 3067) [56]. The dataset was constructed with great attention to and comprehensive details of the ABA biosynthesis gene sequences, with their accession number given in Appendix A. The position of the gene in the cell was obtained according to the BUSCA server (http://busca.biocomp.unibo.it/ (accessed on 22 January 2023)) [57]. The physicochemical properties of ABA biosynthesis proteins were obtained with the ProtParam tool (https://web.expasy.org/protparam/ (accessed on 22 January 2023)), including the number of amino acids, molecular weight, and isoelectric point [58].

### 4.2. Phylogenetic Analysis and Conserved Domain Prediction

For phylogenetic tree reconstruction, the protein sequences were aligned with the MUSCLE application in MEGA 7. Maximum likelihood (ML) analysis was constructed in IQTree v1.6.12 [59] using the best-fit substitution model automatically selected by the software according to the Bayesian information criterion scores and weights (BIC) with partitions [60], and an ultrafast bootstrap (UFB) with 1000 replicates. For conserved structural motifs, the protein sequences were identified using the PFAM website (http://pfam.xfam.org/ (accessed on 22 January 2023)) with default settings. Next, the phylogenetic tree and conserved domain were trimmed and displayed using the EvolView website (https://www.evolgenius.info/evolview/#/ (accessed on 22 January 2023)).

### 4.3. Analysis of ABA Biosynthesis Gene Structure

To study the structural evolution of ABA biosynthesis genes, the regions of exons and introns were predicted in all ABA biosynthesis genes. For this comparative genomic analysis, the genomes and annotation files were retrieved from Phytozome. The ABA biosynthesis gene structures were shown through the Biosequence Structure Illustrator of TBtools v1.106 (https://github.com/CJ-Chen/TBtools/ (accessed on 22 January 2023)) [61].

### 4.4. Chromosome Location and Collinearity Analysis

The chromosome distribution of ABA biosynthesis genes was drawn with TBtools according to the location information from 19 plant genome databases [61]. MCScanx software was used to analyze the collinearity of ABA biosynthesis genes among *P. patens* and *S. caninervis* [26,34], and then the collinearity diagram was obtained through the Multiple Synteny Plot of TBtools.

### 4.5. Gene Expression Analysis of ABA Biosynthesis Genes under Abiotic Stress

Dry *Syntrichia caninervis* Mitt. gametophytes were collected from the Gurbantunggut Desert in Xinjiang, China (44°32′30″ N, 88°6′42″ E). Dry gametophytes were fully hydrated on filter paper saturated with sterile water in glass Petri dishes for 24 h at 25 °C, with light at a photosynthetic photon flux density (PPFD) of 100 μmol/m^2^/s, prior to the abiotic stress treatments, including dehydration, salt, cold, heat and ABA, and then the samples were collected at 0 h, 0.5 h, 1 h, 2 h, 6 h, 12 h, and 24 h in each treatment. For dehydration treatment, the fully hydrated gametophytes were slow dried in a closed desiccator with saturated sodium nitrite at room temperature (67% relative humidity, 25 °C). For salt and ABA treatment, the fully hydrated gametophytes were transferred to new Petri dishes with filter paper saturated with 8 mL 150 mM NaCl or 10 μM ABA solutions at room temperature (30% relative humidity, 25 °C). For cold and heat stress treatments, the fully hydrated gametophytes were placed in Petri plates on water-saturated filter paper and incubated at either 4 °C or 42 °C.

The gametophytes of each treatment at different timepoints were collected and frozen in liquid nitrogen for RNA extraction. Total RNA was isolated using the MiniBEST Plant RNA Extraction Kit (TaKara, Dalian, China) according to the manufacturer’s protocol. The cDNA was synthesized from 1 μg of total RNA using the PrimeScript RT reagent kit (Takara, Dalian, China). Gene expression was performed in 96-well plates with the CFX96 Real-Time PCR Detection System (Bio-Rad, USA), with three technical replicates and two biological replicates [28]. *α-TUB2* was used as an internal control [62], and the relative expression levels of ABA biosynthesis genes (*ScABA1*/*Sc_g04625*, *ScABA4*/*Sc_g04273*, *ScNCED*/*Sc_g07550*, *ScABA2*/*Sc_g02085*, *ScABA3*/*Sc_g05025*, *ScAAO*/*Sc_g15857*) were calculated by the 2^−ΔΔCT^ method [63], and all primers are displayed in Appendix A. Data are expressed as the mean ± SD. Error bars represent standard deviations.

## 5. Conclusions

In this study, we performed a comprehensive analysis of ABA biosynthesis genes, including identification, evolutionary relationships, gene and protein structures, and expression patterns. Our results demonstrated that *S. caninervis* had complete ABA biosynthesis genes, and we first found that chlorophytes had all of these genes. Although the ABA biosynthesis genes of chlorophytes were separated from those of other plant taxa in the phylogenetic tree, the conserved domain was the same in all plants. Furthermore, we found that the ABA biosynthesis gene structures showed huge variation between different evolutionary branches. Noticeably, ABA biosynthesis genes were not located on sex chromosomes and were evenly distributed and dispersed on the other chromosomes. Simultaneously, *ABA1*, *NCED*, and *ABA2* genes had collinearity between the *S. caninervis* and *P. patens* genomes. In *S. caninervis*, all of the ABA biosynthesis genes responded to abiotic stress, and ABA also plays an important role in the stress response. In conclusion, this study not only identified the ABA biosynthesis genes in *S. caninervis* and other representative plants from different plant taxa but also provided sufficient evidence confirming that the ABA biosynthetic pathway was conserved in the evolutionary process.

## Figures and Tables

**Figure 1 plants-12-01114-f001:**
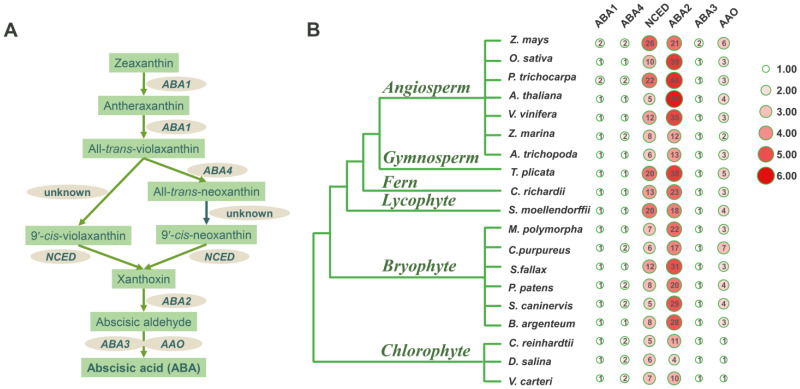
Summarization of ABA biosynthesis genes among different plant taxa. (**A**) The schemes of the ABA biosynthesis pathway. (**B**) The total number of ABA biosynthesis genes in all plants; the green word represents different plant taxa; the number of ABA biosynthesis genes was visualized using TBtools with a homogenization method of log_2_; the circle represents the specific number of ABA biosynthesis genes.

**Figure 2 plants-12-01114-f002:**
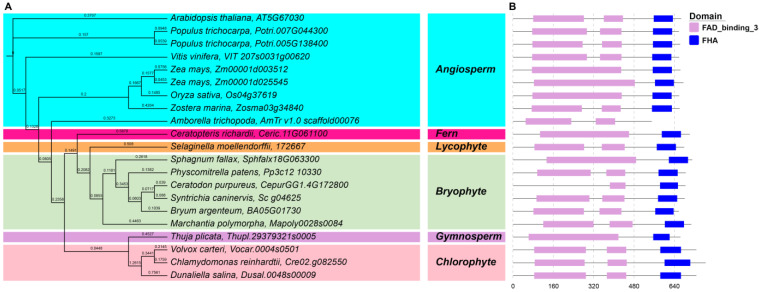
Phylogenetic and deduced protein structure analysis of ABA1. With *Arabidopsis* ABA1 (AT5G67030) as a query, the ABA1 homologous sequences were retrieved from Phytozome using BLASTP. (**A**) The phylogenetic analysis of ABA1; the maximum likelihood (ML) phylogenetic tree was constructed with the full-length amino acid sequence of the *ABA1* genes from 19 species in IQTree v1.6.12 using the best-fit substitution model which was automatically selected by the software; background colors represent different plant taxa. (**B**) The conserved domain analysis of ABA1; conserved motifs were identified by PFAM website, and the motifs are displayed in different colored boxes.

**Figure 3 plants-12-01114-f003:**
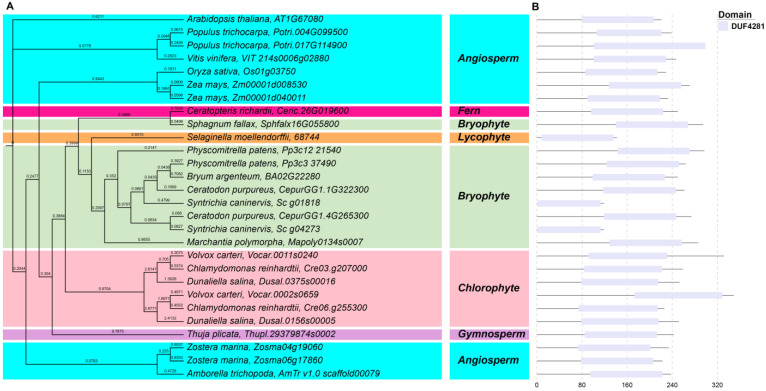
Phylogenetic and deduced protein structure analysis of ABA4. With *Arabidopsis* ABA4 (AT1G67080) as a query, the ABA4 homologous sequences were retrieved from Phytozome using BLASTP. (**A**) The phylogenetic analysis of ABA4; the maximum likelihood (ML) phylogenetic tree was constructed with the full-length amino acid sequence of the *ABA4* genes from 19 species in IQTree v1.6.12 using the best-fit substitution model, which was automatically selected by the software; background colors represent different plant taxa. (**B**) The conserved domain analysis of ABA4; conserved motifs were identified by the PFAM website, and the motifs are displayed in different colored boxes.

**Figure 4 plants-12-01114-f004:**
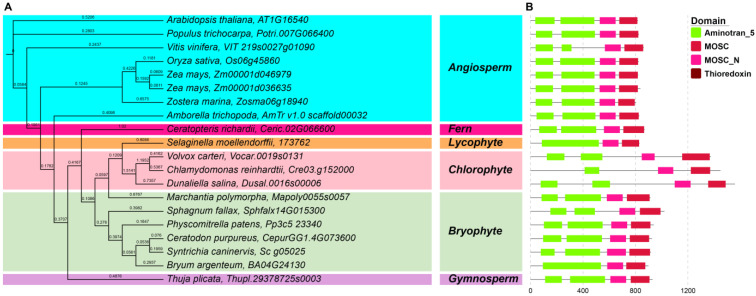
Phylogenetic and deduced protein structure analysis of ABA3. With *Arabidopsis* ABA3 (AT1G16540) as a query, the ABA3 homologous sequences were retrieved from Phytozome using BLASTP. (**A**) The phylogenetic analysis of ABA3; the maximum likelihood (ML) phylogenetic tree was constructed with the full-length amino acid sequence of the *ABA3* genes from 19 species in IQTree v1.6.12 using the best-fit substitution model, which was automatically selected by the software; background colors represent different plant taxa. (**B**) The conserved domain analysis of ABA3; conserved motifs were identified by the PFAM website, and the motifs are displayed in different colored boxes.

**Figure 5 plants-12-01114-f005:**
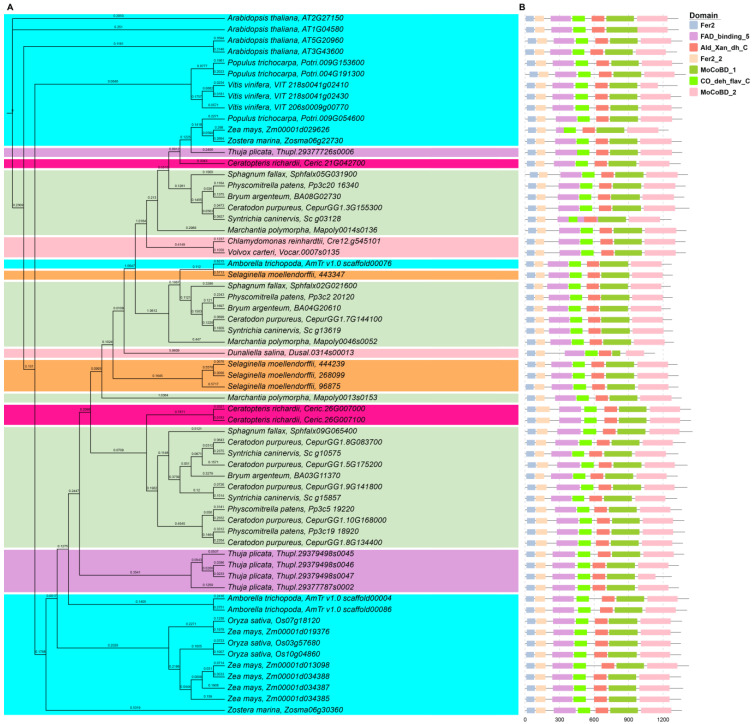
Phylogenetic and deduced protein structure analysis of AAO. With *Arabidopsis* AAO (AT2G27150) as a query, the AAO homologous sequences were retrieved from Phytozome using BLASTP. (**A**) The phylogenetic analysis of AAO; the maximum likelihood (ML) phylogenetic tree was constructed with the full-length amino acid sequence of the *AAO* genes from 19 species in IQTree v1.6.12 using the best-fit substitution model, which was automatically selected by the software; background colors represent different plant taxa. (**B**) The conserved domain analysis of AAO; conserved motifs were identified by the PFAM website, and the motifs are displayed in different colored boxes.

**Figure 6 plants-12-01114-f006:**
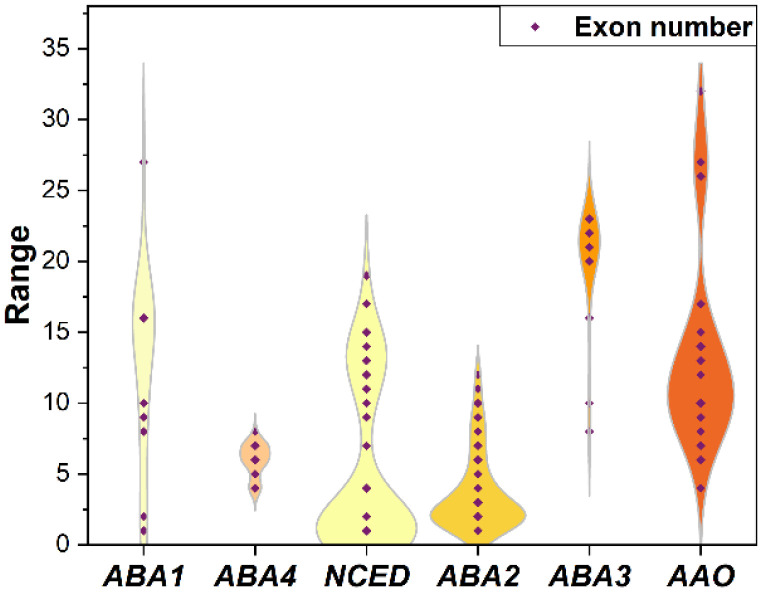
Exon number comparison of ABA biosynthesis genes in 19 plant species. TBtools analyzed the exon-intron structure of ABA biosynthesis genes from 19 species with default parameters; exon numbers are displayed in the violin plot.

**Figure 7 plants-12-01114-f007:**
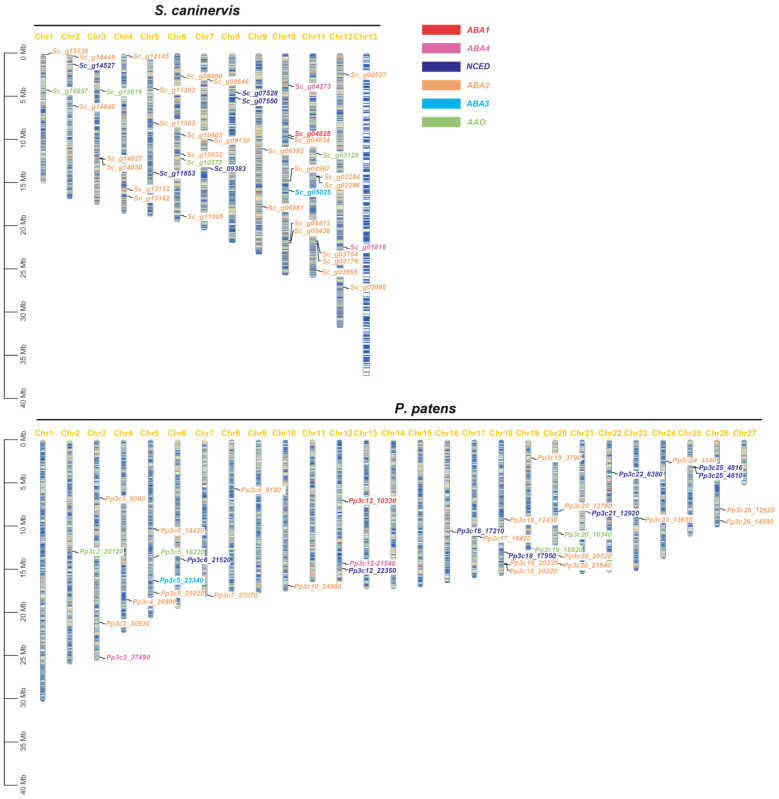
Chromosome distribution of ABA biosynthesis genes in *S. caninervis* and *P. patens*. The bryophyte species name is given at the top of the graphic, and the chromosome name is given at the top of each bar. The vertical scale on the left shows the size of the chromosome, and the black lines indicate the corresponding positions of genes. The scale of the chromosomes is millions of base pairs (Mb).

**Figure 8 plants-12-01114-f008:**
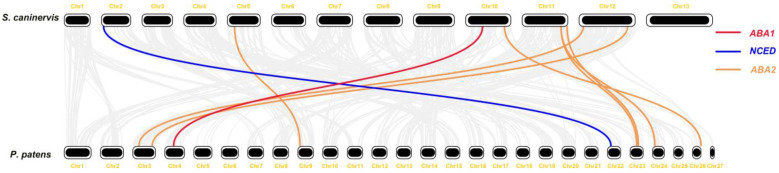
Collinear analysis of ABA biosynthesis in *S. caninervis* and *P. patens*. Gray lines in the background show the collinear relationship of the whole genome, while the colored line mainly shows the collinear ABA biosynthesis gene pair.

**Figure 9 plants-12-01114-f009:**
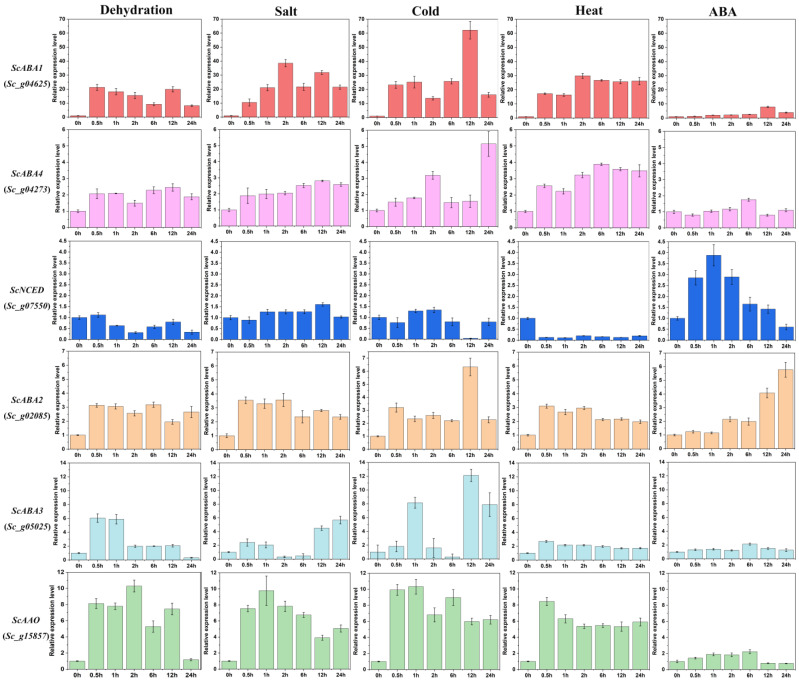
Relative transcript abundance of ABA biosynthesis genes in *S. caninervis* under different stress treatments. Error bars represent the SD of three biological repeats. Relative expression values were obtained from 2^−ΔΔCt^ comparing different stress times with control (0 h), respectively.

**Table 1 plants-12-01114-t001:** Characteristics of ABA biosynthesis genes in *S. caninervis* and *P. patens*.

Species	Gene Name	Gene ID	Protein Length (aa)	MW (kDa)	pI	Subcellular Location
*S. caninervis*	*ScABA1*	*Sc_g04625*	681	73.9	8.15	chloroplast
	*ScABA4*	*Sc_g04273*	252	27.8	9.71	chloroplast
		*Sc_g01818*	199	13.6	7.66	endomembrane
	*ScNCED*	*Sc_g09383*	615	68.2	7.57	chloroplast
		*Sc_g07550*	590	64.8	5.97	chloroplast
		*Sc_g14527*	540	60.5	5.59	cytoplasm
		*Sc_g11853*	594	66.3	5.74	chloroplast
		*Sc_g07528*	559	62.8	6.30	cytoplasm
	*ScABA2*	*Sc_g02085*	317	33.6	8.24	mitochondrion
		*Sc_g06861*	305	31.6	6.71	mitochondrion
		*Sc_g06383*	254	26.1	5.78	mitochondrion
		*Sc_g10403*	335	36.7	7.16	mitochondrion
		*Sc_g11005*	300	32.5	5.52	cytoplasm
	*ScABA3*	*Sc_g05025*	919	101.8	7.10	extracellular space
	*ScAAO*	*Sc_g15857*	1323	143.7	6.30	chloroplast
		*Sc_g10575*	1334	144.9	6.25	cytoplasm
		*Sc_g13619*	1287	138.9	6.67	cytoplasm
		*Sc_g03128*	1273	138.5	6.14	cytoplasm
*P. patens*	*PpABA1*	*Pp3c12_10330*	685	74.4	7.85	chloroplast
	*PpABA4*	*Pp3c12_21540*	297	33.0	10.03	mitochondrion
		*Pp3c3_37490*	264	29.2	9.69	chloroplast
	*PpNCED*	*Pp3c25_4810*	585	65.0	5.84	cytoplasm
		*Pp3c25_4816*	585	65.0	5.84	cytoplasm
		*Pp3c16_17210*	585	64.8	5.63	cytoplasm
		*Pp3c22_6380*	538	61.2	5.78	cytoplasm
		*Pp3c21_12920*	622	69.9	5.74	chloroplast
	*PpABA2*	*Pp3c8_9180*	329	35.6	8.77	chloroplast
		*Pp3c3_30930*	316	33.6	8.41	mitochondrion
		*Pp3c17_16420*	306	31.7	6.60	mitochondrion
		*Pp3c5_14430*	359	39.3	5.89	cytoplasm
		*Pp3c18_12430*	349	38.0	8.36	chloroplast
	*PpABA3*	*Pp3c5_23340*	940	104.2	6.34	chloroplast
	*PpAAO*	*Pp3c5_19220*	1363	147.6	5.86	chloroplast
		*Pp3c19_18920*	1391	151.0	6.46	cytoplasm
		*Pp3c2_20120*	1283	139.2	6.46	cytoplasm
		*Pp3c20_16340*	1396	152.2	6.11	cytoplasm

## Data Availability

All of the accession numbers of ABA biosynthesis genes in this study can be found in Appendix A.

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
