# Peer review of "Genome-Wide Characterization and Expression Profiling of ABA Biosynthesis Genes in a Desert Moss *Syntrichia caninervis"

_plants, 2023, doi:10.3390/plants12051114_

Round 1

Reviewer 1 Report

The manuscript describes the identification of genes related to ABA biosynthesis in the desert moss species Syntrichia caninervis, as had previously been done in the model bryophyte Physcomitrella patens (Minami et al., 2003, 2005) also showing that this species is able to produce ABA when subjected to hyperosmotic stress induced by mannitol treatment. In this work, several genes involved in all steps of ABA biosynthesis from xanthophylls have been identified in S. caninervis showing high homology with plant genes. Some of these genes might be functional, some others not. This is probably the main concern on this work (as already highlighted by authors), it is not clear whether all the genes identified are functional or simply share homology with those found in plants (for most of which there is a functional characterization). This is of special relevance for NCED, as the main bottleneck in ABA biosynthesis; actually, gene expression of the chosen entry (Sc_g07550) does not match the expected outcome for some of the treatments (no upregulation in most of the cases except for ABA treatment, as it should be expected the opposite). Moreover, there is no data confirming that S. caninervis actually produces ABA under the conditions assayed, the analysis of the phytohormone could also contribute to confirm the findings presented in this work. 

An additional issue is the fact that no ABA receptors have been so far identified (recently a glutamate receptor-like gene was identified in P. patens that could be involved in ABA signal perception), but no PYR/PYL homologues, as far as I know. This could further strengthen the role of ABA as an endogenous regulator in moss species, and there is no reason to believe that ABA signal transduction is channeled through a different pathway in bryophytes.

Moreover, stress and ABA treatments should be better explained and provide other phenotype/biochemical/physiological measurements that confirm the efficiency of treatments.

Author Response

Dear Reviewer:

We would like to thank you for taking your time to review our manuscript titled “Genome-Wide Characterization and Expression Profiling of ABA biosynthesis genes in a desert moss Syntrichia caninervis” plants-2154888, and providing many helpful suggestions and commends, which will all prove invaluable in the revision and improvement of our paper, as well as in guiding our research in the future.

We have studied your comments point by point, please see the attachment. We have modified the language in MDPI “Language Editing Services”.

We hope that the revised version of the manuscript is now acceptable for publication in Plants. Thank you again for your valuable comments and suggestions.

Yours sincerely,

Xiujin Liu

Reviewer 2 Report

See comments on attached file.

Author Response

Dear Reviewer:

We would like to thank you for taking your time to review our manuscript titled “Genome-Wide Characterization and Expression Profiling of ABA biosynthesis genes in a desert moss Syntrichia caninervis” plants-2154888, and providing many helpful suggestions and commends, which will all prove invaluable in the revision and improvement of our paper, as well as in guiding our research in the future.

We have studied your comments point by point, please see the attachment. Moreover, we have modified the language in MDPI “Language Editing Services”.

We hope that the revised version of the manuscript is now acceptable for publication in Plants. Thank you again for your valuable comments and suggestions.

Yours sincerely,

Xiujin Liu

Round 2

Reviewer 1 Report

I agree with authors that the gene they highlight is likely not involved in ABA biosynthesis, therefore, I suggest authors perform a more detailed screening of NCED-homologous genes and confirm their potential involvement in ABA biosynthesis. As it stands, its role as ABA biosynthetic gene is highly speculative.

ABA levels in plants subjected to the different treatments must be provided to correlate with gene expression.

I agree that it might be conserved in different clades, but authors did not respond to my question...what about ABA receptors?

Author Response

Dear Reviewer1:

We really appreciate you for your kindly guidance, it will all prove invaluable in the revision and improvement of our paper, as well as in guiding our research in the future. In accordance with your suggestions, we have studied your comments point by point. All authors have approved the response and the revised version of the manuscript. Please see the attachment.

We hope that the response is now acceptable for publication in Plants. If you have any queries, please do not hesitate to contact me.

Thank you again for your valuable comments and suggestions. I look forward to hearing from you soon in due course.

Yours sincerely,

Xiujin Liu

Xinjiang Institute of Ecology and Geography,

Chinese Academy of Sciences

E-mail: liuxiujin@ms.xjb.ac.cn

Reviewer 2 Report

Overall, the authors have taken in consideration most of the comments. However, there are still improvements that add scientific soundness to this objective. See below

1. Authors have provided two versions of the phylogenetic trees i.e., NJ and ML. This means that the authors are not clear in choosing the most appropriate method to infer the phylogeny. Since they are discussing the genes in terms of their evolution, they must adapt ML approach. 

(See here some guide points about ML Tree)

Why do we want to use ML and not the other methods?

If we need a least biased tree.

If we need a tree with the smallest variance and most robust to violations of model assumptions.

A tree that should outperform all others on statistical bases.

It is better to use NJ as a quick look and a to-begin-with tree but an ML tree with the best model is what you ll need at the end.

2. I had suggested authors to use ModelFinder, which they tried, however, they did not add the Substitution MODEL which was chosen by the software for the generation of the ML trees. Authors are advised to include the model e.g., was it JTT matrix based model or JTT model or something else. in the IQTree results, there is always mention of the most appropriate model adapted by the software and statistical bases of choosing the model. It must be available in the third panel (FULL RESULT) after your job is successful.

3. Authors did not take into consideration the comment about gene loss events. This analysis is important as it add useful information regarding gene loss in terms of evolution and authors can confirm if their hypothesis about the evolution is legit or not. For such analyses, they can use Notung 2.9 (http://www.cs.cmu.edu/~durand/Notung/) or similar software. 

Author Response

Thank you for reviewing our manuscript (Genome-Wide Characterization and Expression Profiling of ABA Biosynthesis Genes in a Desert Moss Syntrichia caninervis, plants-2154888) and offering valuable advice. We really appreciate you for your kindly guidance on our bioinformatic analysis methods, and offering many practical and free website. It will all prove invaluable in the revision and improvement of our paper, as well as in guiding our research in the future. We have studied your comments point by point, revised the manuscript accordingly. Please see the attachment.

Last time, we have modified the language in MDPI “Language Editing Services”, and uploaded the confirmation certificate.

We hope that the revised version of the manuscript is now acceptable for publication in Plants. If you have any queries, please do not hesitate to contact me.

Thank you again for your valuable comments and suggestions. I look forward to hearing from you soon in due course.

Yours sincerely,

Xiujin Liu

Xinjiang Institute of Ecology and Geography,

Chinese Academy of Sciences

E-mail: liuxiujin@ms.xjb.ac.cn

Round 3

Reviewer 2 Report

CURRENT VERSION IS FINE AND AUTHORS HAVE EITHER ADAPTED THE SUGGESTIONS OR MODIFIED THEIR TEXT.

Author Response

Dear Reviewer2:

Thank you again for your review and commands. We really appreciate you for your kindly and multiple guidance, it will help our research in the future. We have proofread the entire text, modified punctuation, replaced some words such as changing “one huge gene family” to “a multigene family” in the third revised manuscript. Please see the manuscript, which was named as ”Manuscript-revised-reviewer2 (Round 3)”. All authors have approved the response and the revised version of the manuscript.

We hope that the revised version of the manuscript is now acceptable for publication in Plants. If you have any queries, please do not hesitate to contact me.

Thank you again for your valuable comments and suggestions. I look forward to hearing from you soon in due course.

Yours sincerely,

Xiujin Liu

Xinjiang Institute of Ecology and Geography,

Chinese Academy of Sciences

E-mail: liuxiujin@ms.xjb.ac.cn